# Assessing Mental Pain as a Predictive Factor of Suicide Risk in a Clinical Sample of Patients with Psychiatric Disorders

**DOI:** 10.3390/bs12040111

**Published:** 2022-04-16

**Authors:** Marta Ielmini, Giulia Lucca, Eric Trabucchi, Gian Luca Aspesi, Alessandro Bellini, Ivano Caselli, Camilla Callegari

**Affiliations:** 1Department of Medicine and Surgery, Division of Psychiatry, University of Insubria, 21100 Varese, Italy; marta.ielmini@uninsubria.it (M.I.); giulia1lucca@gmail.com (G.L.); trabucchieric@gmail.com (E.T.); gian.aspesi@gmail.com (G.L.A.); ivano.caselli@uninsubria.it (I.C.); 2Department of Medicine and Surgery, Division of Psychiatry, University of Pavia, 21100 Pavia, Italy; alessandro.bellini1992@gmail.com

**Keywords:** suicide risk, prevention, mental pain, suicidality, psychache

## Abstract

According to contemporary suicidology, mental pain represents one of the main suicide risk factors, along with more traditional constructs such as depression, anxiety and hopelessness. This work aims to investigate the relationship between the levels of mental pain and the risk to carry out suicide or suicide attempt in the short term in order to understand if a measurement of mental pain can be used as a screening tool for prevention. For this purpose, 105 outpatients with psychiatric diagnosis were recruited at the university hospital of Varese during a check-up visit and were assessed by using psychometric scales of mental pain levels, hopelessness, anxiety and depression. Clinical and sociodemographic variables of the sample were also collected. A period of 18 months following the recruitment was observed to evaluate any suicides or attempted suicides. Subjects numbering 11 out of 105 committed an attempted suicide. From statistical analyses, high values of the Beck Depression Inventory (BDI-II), Mental Pain Questionnaire (OMMP) and Hamilton Rating Scale for Depression (HAM-D) scales showed a significant association with the risk of carrying out a suicide attempt and, among these, OMMP and BDI-II showed characteristics of good applicability and predictivity proving suitable to be used as potential tools for screening and primary prevention of suicidal behavior.

## 1. Introduction

Suicide is a major public health problem as it is a leading cause of death. It is estimated that there are nearly one million victims of suicide globally every year and that attempted suicides (AS) number up to 30 times more frequent than suicides [1,2]. According to contemporary suicidology, the major risk factor for suicide is not represented by psychiatric diagnosis but by the levels of psychological pain (“psychache”), described as an intolerable subjective experience that the person wishes to end by any means available, including physical death, and conceptualized as a perception of negative changes in the self and its functions that are accompanied by negative feelings [3,4,5,6,7]. Several case–control studies comparing patients suffering from psychiatric disorders versus persons without a psychiatric diagnosis have highlighted a relationship between a history of suicide attempts and psychological pain, concluding that psychological pain is significantly correlated with both previous suicides and suicide attempts, and with the presence of suicidal ideation [8,9,10,11]. In recent years, there has been increasing attention on the role of individual factors and, among those, to mental pain in adverse life event responses [12,13] independently from psychiatric diagnosis and/or depressive symptoms [4,14]. The existing studies on this topic are mainly cross-sectional studies [8,15,16]. This model allows the identification of the correlation between levels of mental pain and a history of previous suicides and suicide attempts, or between levels of mental pain and current suicidal ideation, but it does not allow an understanding of whether the identified potential risk could translate results into a concrete attempt. Furthermore, most papers refer to samples of patients with mood disorders, leaving out many of the diagnoses historically associated with suicide risk (e.g., personality disorders and substance abuse) and, thus, deviating from the clinical reality of psychiatric services [17,18]. The few prospective works conducted on this subject show that the high levels of mental pain predict changes in suicidal ideation both in the short term (weeks and months) and in the long term (up to four years); however, these works [9,19,20,21] were conducted on non-clinical samples (university students) and evaluated changes in ideation but not actual complete suicides or suicidal attempts in the follow-up period. Starting from the assumption that mental pain is a measurable construct by using psychometric scales, this study aims to identify if there are any correlations between mental pain levels and suicide or suicide attempts in the short term and, if so, to identify a psychometric tool in the context of suicide prevention [16].

## 2. Materials and Methods

This is a longitudinal study: 105 patients were recruited between January 2019 and December 2019 by administering evaluation scales related to the levels of depression, anxiety, hopelessness and mental pain, aiming at the following endpoints: to calculate any correlation between the level of mental pain, measured using the scales administered at the time of recruitment, and the risk of suicide or attempted suicide in the following 18-month period; to deepen the correlation between socio-demographic, clinical and psychometric variables with suicide risks in the short term in order to help clinicians identify candidates for a targeted prevention pathway. Clinical and sociodemographic variables of the sample were also collected.

The patients were recruited at the university hospital of Varese (Azienda Socio-Sanitaria Territoriale Sette Laghi) on the occasion of a check-up visit at the psychiatric outpatients care services of Varese, Azzate and Arcisate, and at the outpatients clinic for anxiety and depression fulfilling the following inclusion criteria: over 18 years of age; presence of a psychiatric illness diagnosed following the diagnostic criteria of the *Statistical Diagnostic Manual of Mental Disorders, Fifth Edition* (DSM-5); and signing of an informed consent for the use of data anonymously for research purposes. Minor age and the presence of a cognitive or linguistic barrier that compromised the understanding of the study were considered as the exclusion criteria. All personally sensitive information contained in the database used for this study was previously de-identified according to the Italian legislation (D.L. 196/2003, art. 110, —4 July 2008 art. 13). Since data were made anonymous and unidentifiable, the Provincial Health Ethical Review Board (Ethics Committee of Insubria—*Azienda Socio Sanitaria Sette Laghi, Varese, Italy*) consulted prior to the beginning of the study confirmed that, since the study consists in a longitudinal investigation, it does not require an approval process from the board. The study was carried out in accordance with the Declaration of Helsinki (with amendments) and Good Clinical Practice.

Subjects suffering from major neuropsychiatric pathologies, such as epilepsy, intellectual disabilities or genetic syndromes with psychiatric correlates, and patients suffering from conditions that did not allow the completion of the evaluation, such as linguistic problems, severe dyslexia and poor knowledge of the Italian language, were excluded from the study.

During the first assessment, the following psychometric scales were administered:-**Suicide Columbia Severity Rating Scale (C-SSRS)** [22]: Scale administered by the clinicians that investigates the presence of a suicidal ideation, the intensity of the ideation and the behavior and lethality of the suicidal act;-**Hamilton Rating Scale for Anxiety (Ham-A)** [23]: Scale administered by the clinician composed of 14 elements that investigates anxiety symptoms, both psychological and somatic;-**Hamilton Rating Scale for Depression (Ham-D)** [24]: Scale administered by the clinicians; it evaluates the presence and severity of psychic and psychosomatic depressive symptoms, consisting of 21 items;-**Psychache Scale (PAS)** [25]: Self-administered scale composed of 13 items; measures the frequency and intensity of mental pain;-**Mental Pain Questionnaire (OMMP)** [14]: Self-administered scale that measures the levels of mental pain, consisting of forty-four items that investigate the nine main components of mental pain;-**Beck Depression Inventory-II (BDI II)** [26]: Self-administered scale that estimates the presence and intensity of depressive symptoms referred to the last two weeks by using 21 guided answer questions;-**Childhood Trauma Questionnaire (CTQ)** [27,28]: Self-administered questionnaire consisting of 28 items; it evaluates different types of child maltreatment, such as physical, sexual, emotional abuse and both physical and emotional neglect;-**Beck Hopelessness Scale (BHS)** [29]: Self-administered scale that evaluates the three fundamental aspects of a construct called hopelessness by using 20 statements with a “true” or “false” responses.

Clinicians were trained to administer the psychometric scales. 

After 18 months from the date of recruitment, the hospital (PORTALE) and territorial (PSYCHE) databases and medical records were analyzed with the aim of underlining any suicides or attempted suicides that may have occured during the follow-up period. PORTALE is a management software used within ASST Sette Laghi. It has been active since 2008 and is accessible only by clinicians or medical residents by using a personal username and password.

PSYCHE is the other database used; it is available for consultation using a special card owned only by doctors. Complete suicides were considered, or suicidal attempts that led to access to the emergency room requiring observation for at least 24 h, or subsequent hospitalization, or events that were followed by an emergency therapeutic intervention. Attempts that did not lead to accessing emergency services but that were described as objectively anti-conservative by the referring clinicians and reported in the patient’s clinical documentation were also considered.

Statistical analyses were performed using the IBM^®^ SPSS^®^ Statistics Version 24 statistical package. The sample size was calculated based on previous investigation [8]. 

The variables investigated were presented as the mean and standard deviation for continuous variables and as the mode and percentages for qualitative variables. The associations were evaluated using parametric tests in the case of qualitative variables with position indices in the range between −1 and 1, while the quantitative variables were evaluated with Kolmogorov–Smirnov test, which reports *p* < 0.05.

## 3. Results

The sample consists of 105 outpatients, 74 males and 31 females. Most patients are in the age group between 50 and 59 years (23.8%); the age of onset of the disease is between 10 and 19 years in 31.4% of cases; 74.3% of the sample is in polypharmacotherapy, including different categories (i.e., antidepressant and mood stabilizer or antidepressant and anxiolytics); 68.6% of the sample is in treatment with benzodiazepines. The most represented psychiatric diagnoses are as follows: Major Depressive Disorder, Personality Disorder and Adjustment Disorder. During the follow-up period, complete suicides or suicidal attempts were considered. ASs were included when access to the emergency room requiring observation for at least 24 h was involved, or subsequent hospitalization or which were followed by an emergency therapeutic intervention. Attempts that did not lead to access to emergency services but that were described as objectively anti-conservative by the referring clinicians and reported in the patient’s clinical documentation were also considered. During the period of observation, 11 out of 105 subjects attempted suicide (AS). Among the AS group, four patients had drugs and alcohol abuse, two subjects had cut themselves and one patient arrived at the emergency service after defenestration.

The variables that showed a statistically significant prevalence in the group of patients with new suicide attempts are shown in Table 1. The invalidity subcategory was found to be the one most at risk of having a new AS (*p* = 0.002). The absence of comorbidities is associated with a lower suicide risk, while substance abuse disorder is more frequent in those who have made a suicide attempt in the follow-up, particularly regarding alcohol abuse. The presence of encephalopathy is also more represented in the group of patients who have made a suicide attempt during the follow-up. The use of a polypharmacotherapy, which also included antipsychotics, was found with a higher prevalence in subjects with new suicide attempts. Lastly, the presence of a previous suicide attempt in the medical history represents one of the main anamnestic risk factors.

Subjects who attempted suicide during the follow-up period show higher scores on all the psychometric scales; however, only the BDI-II, PAS, OMMP and HAM-D scales’ scores achieved statistical significance.

Figure 1 and Table 2 show the averages of the scales’ scores, considering the coefficients of statistical significance.

Using binary logistic regression, the predictive models of suicide risk were searched for and are reported in Table 3; high mean scores obtained on BDI-II, HAM-D and OMMP scales increased the likelihood of suicide attempts in the following 18 months. In particular, the increase in one unit in the total score on the BDI-II scale translates into a probability greater than 5.6% of attempting a new suicide (*p* = 0.020); similarly, increases in the HAM-D score (*p* = 0.028) by one unit corresponded to a 12.9% greater risk of incurring a new AS. Finally, an increase in one unit in the OMMP scale score (*p* = 0.048) results in a probability of making a new AS greater than 10.9%.

## 4. Discussion

The socio-demographic and clinical risk factors identified by this study confirm the existing literature data. The results show that people with invalidity are more exposed to attempt a new AS (*p* = 0.002), underling the role of the social support and of occupation as protective toward suicide risks, as reported in the literature. With respect to anamnestic risk factors, we observed results that were in line with the literature: People with a history of drug abuse or an ongoing problem with substances have a higher risk for committing AS (both *p* = 0.00), and individuals that previously tried to commit suicide also had higher risks (*p* = 0.001). Moreover, clinical comorbidities and more severe clinical problems resulted in greater risk of new suicides (*p* = 0.00), as already described by other authors [30]. In particular, among the comorbidities, we found that encephalopathy was the most frequently associated factor. Encephalopathy has been widely studied as a risk factor for AS or suicide, but the existing results seem contrast this observation [31]. These data could be explained by the emotional burden associated with these conditions. Increasing evidence supports the role of genetic and epigenetic factors associated with the social context and comorbidities in vulnerability relative to suicidal risks and seem to increase other psychiatric conditions [12,32,33,34]. Invalidity, drug and alcohol abuse and previous AS seem to be the most related risk factors that are significantly more present in the group of patients that experimented with new suicide attempts (respectively, *p* = 0.00, *p* = 0.000; *p* = 0.010). It is noted that people with alcohol abuse (*p*-value: 0.000) have the highest risk. It is reasonable to assume that alcohol abuse can precipitate suicidal behavior through disinhibition, impulsiveness and a lack of judgment related to the use of this substance; nevertheless, alcohol abuse is capable of exacerbating depressive and psychotic symptoms that play a negative role also at a social level, leading the subject to marginalization, poverty and to the development of further somatic complications [35]. The results emerging from the analysis of psychometric scales used to measure the levels of mental pain, anxiety and depression, hopelessness and their relationship with the risk of attempting a new suicide are very interesting. Indeed, the mean values of the total scores measured on the BDI-II, PAS, OMMP and HAM-D scales were significantly higher in subjects who had acted out a new AS within 18 months, while the scores obtained on the other scales did not reach a statistical significance capable of correlating to the same risk. This allows the confirmation that the levels of mental pain, depression and hopelessness are higher in those who will perform a new AS, as already found in other studies [14,24,36,37,38,39,40], and that these psychometric tools are potentially useful in discriminating between “high suicidal risk” and “low suicidal risk”. In order to identify the psychometric scales that can be used as a suicide risk screening tool, the reliability of each of these to constitute a predictive model was analyzed. The BDI-II, HAM-D and OMMP scales seem to be able to estimate the probability of incurring a new AS. Among these, BDI-II and OMMP are self-reported scales and, therefore, are much easier to administer than HAM-D, which is operator-dependent [41] It is also important to underline that these two scales measure two different constructs: The BDI-II scale measures the levels of depression, so it could be limited only to this type of diagnosis. In our sample, although several diagnoses were included, the diagnoses of depressive disorder and adjustment disorder with depressive symptoms are the most represented; this fact, together with the limited sample size, they could represent a confounding factor. The OMMP scale, by measuring an internal construct such as mental pain, turns out to be diagnosis-independent. This provides this tool with the characteristic of being potentially applicable to multiple diagnoses and multiple contexts, including extra-psychiatric ones, and this last point is, in our opinion, worthy of further study. Since the COVID-19 pandemic period happened during the 18 months follow-up period for the entire sample, it has not been considered as a variable. Despite the evidence of increased suicide risk and addictive behaviors during the COVID-19 pandemic [42,43], the purpose of the study is to analyze the internal factor of mental pain rather than external factors. The study presents some limits such as the small sample size, but it shows interesting results that confirm the existent literature and suggests useful tools for routine clinical practices.

## 5. Conclusions

The results showed that high levels of depression, measured with the HAM-D and BDI-II scales, and high mental pain levels, measured with the OMMP scale, are correlated with a greater probability of incurring a suicide attempt in the short term. Furthermore, thanks to the characteristics of rapidity and the simplicity of use of these individual scales, it is considered useful to include BDI-II and the OMMP among the screening methods; these two scales can be used to improve suicide prevention, representing a useful tool for early identifying patients with increased suicide risks for clinicians. In suicide prevention, some authors highlight the effect of combined therapy as a collaboration between multidisciplinary teams having a positive impact on the patients’ mental state [44,45]. 

## Figures and Tables

**Figure 1 behavsci-12-00111-f001:**
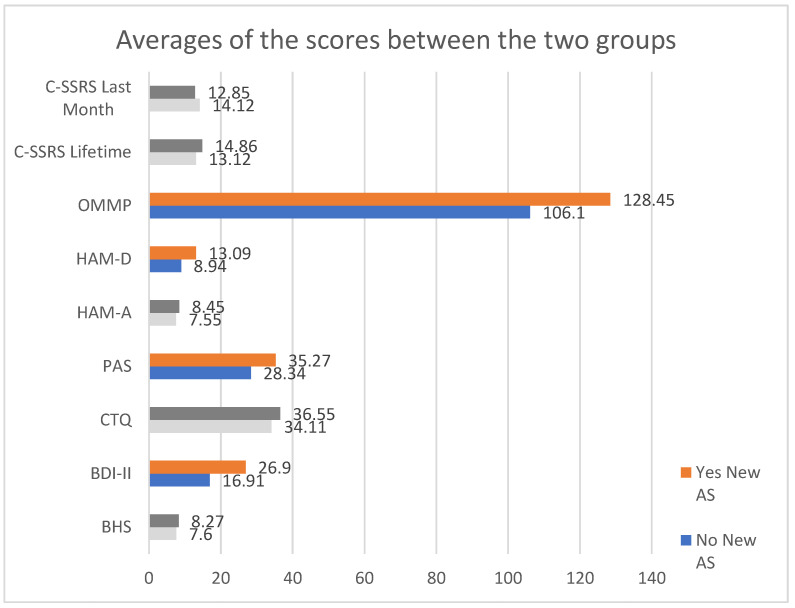
Averages of the psychometric scales’ scores associated with the risk of new AS. Grayed out—the insignificant scores; colored out—the scores with *p* value < 0.05.

**Table 1 behavsci-12-00111-t001:** Socio-demographic and clinical variables that showed a significant difference in the two groups at follow-up.

Employment	No New AS (%)	New AS (%)	*p* Value
Invalidity	3.19%	27.27%	0.002
**Comorbidity**			
None	74.46%	18.18%	0.000
Substance abuse disorder	5.32%	36.36%	0.000
Encephalopathy	1.06%	18.18%	0.000
**Antipsychotic treatment**			
None	65.96%	18.18%	0.006
Quetiapine	8.51%	45.45%	0.006
**History of drug abuse**			
None	77.66%	36.36%	0.008
Substance Poliabuse	9.57%	36.36%	0.008
**Current drug abuse**			
None	82.98%	27.27%	0.000
Alcol	7.44%	45.45%	0.000
Previous AS			
No	67.02%	27.27%	0.010
Yes	32.98%	72.73%	0.010

**Table 2 behavsci-12-00111-t002:** Averages of psychometric scales’ scores associated with the risk of new AS and differences between the two groups.

Scales	Averages NO New AS Group	Averages YES New AS Group	Difference of Averages	*p* Value
BHS	7.60	8.27	0.67	0.626
**BDI-II**	16.91	26.90	9.99	**0.018**
CTQ	34.11	36.55	2.44	0.509
**PAS**	28.34	35.27	6.93	**0.035**
HAM-A	7.55	8.45	0.9	0.721
**HAM-D**	8.94	13.09	4.14	**0.033**
**OMMP**	106.10	128.45	22.34	**0.023**
C-SSRS Lifetime	13.12	14.86	1.74	0.362
C-SSRS Last Month	14.12	12.85	1.27	0.240

**Table 3 behavsci-12-00111-t003:** Association between psychometric scales and predictive risk of incurring a new AS.

Scales	R^2^ Nagelkerke	*p* Value	Exp (B)
C-SSRS Lifetime	0.022	0.431	1.062
C-SSRS Last Month	0.021	0.506	0.937
BHS	0.003	0.686	1.024
**BDI-II**	0.104	**0.020**	1.056
CTQ	0.028	0.222	1.060
PAS	0.058	0.080	1.044
HAM-A	0.007	0.544	1.040
**HAM-D**	0.095	**0.028**	1.129
**OMMP**	0.078	**0.048**	1.109

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
