# Peer review of "Assessing Mental Pain as a Predictive Factor of Suicide Risk in a Clinical Sample of Patients with Psychiatric Disorders"

_behavsci, 2022, doi:10.3390/bs12040111_

Round 1

Reviewer 1 Report

The presented article deals with the topic Assessing mental pain as a predictive factor of suicide risk in a clinical sample of patients with psychiatric disorders. I consider the chosen topic to be very topical and problematic in view of the fact that the number of cases of suicidal behavior in psychiatric patients is increasing.

Therefore, I believe that the article is beneficial and suitable for publication in the journal. However, I have a few remarks about the submitted article that need to be added. This will give the article more quality.

When reading the article, the intention of the authors is obvious. The theoretical insight is therefore relatively well described. The authors use a sufficient amount of relevant resources, which I also consider beneficial. The methodology and survey are appropriately described together with the presentation of the results. When presenting the results, however, I could imagine a more detailed description, or directly a model case from practice. What types of suicides are happening? How can they be prevented?

These are the questions that I think are appropriate to answer in the conclusions chapter. It is in this chapter that I recommend mentioning, for example, the effect of combined therapy, as a collaboration between multidisciplinary teams that apply and have a positive impact on the patient's mental state. One such example article may be e.g.

or

The essence of combined therapy is evident from the given articles, I recommend to add a certain research to the conclusions, which will show that combined therapy has its fixed place in cooperation on the given issue.

In any case, this is a minor adjustment and rather a recommendation. I believe that such articles should have a broader starting point in the weld and point to interdisciplinary and multidisciplinary cooperation. Which is the essence of comprehensive rehabilitation not only in neurological patients, but also in psychiatric patients.

Author Response

Reviewer 1

The presented article deals with the topic Assessing mental pain as a predictive factor of suicide risk in a clinical sample of patients with psychiatric disorders. I consider the chosen topic to be very topical and problematic in view of the fact that the number of cases of suicidal behavior in psychiatric patients is increasing.

Therefore, I believe that the article is beneficial and suitable for publication in the journal. However, I have a few remarks about the submitted article that need to be added. This will give the article more quality.

When reading the article, the intention of the authors is obvious. The theoretical insight is therefore relatively well described. The authors use a sufficient amount of relevant resources, which I also consider beneficial. The methodology and survey are appropriately described together with the presentation of the results. When presenting the results, however, I could imagine a more detailed description, or directly a model case from practice. What types of suicides are happening? How can they be prevented?

These are the questions that I think are appropriate to answer in the conclusions chapter. It is in this chapter that I recommend mentioning, for example, the effect of combined therapy, as a collaboration between multidisciplinary teams that apply and have a positive impact on the patient's mental state. One such example article may be e.g.

or

The essence of combined therapy is evident from the given articles, I recommend to add a certain research to the conclusions, which will show that combined therapy has its fixed place in cooperation on the given issue.

In any case, this is a minor adjustment and rather a recommendation. I believe that such articles should have a broader starting point in the weld and point to interdisciplinary and multidisciplinary cooperation. Which is the essence of comprehensive rehabilitation not only in neurological patients, but also in psychiatric patients.

Thank you for your positive comments. Regarding the presentation of the results, a more detailed description of the types of suicide has been provided. We added the paragraph on combined therapy with reference to the suggested articles.

Reviewer 2 Report

This manuscript provides preliminary data to the potential use of Beck Depression Inventory (BDI-II) and Mental Pain Questionnaire (OMMP) as screening tools of suicide risk. However, the manuscript would benefit of major changes.

Background:

  1. Line 28-29: with regard to the estimation of victims of suicide I would suggest to consider also World Health Organization. Suicide worldwide in 2019. Global Health Estimates, JUNE 2021.
  2. Line 32: I would suggest authors to better describe dimensions related to psychache (e.g. guilt, loneliness, fear, irreversibility of pain, emotional flooding, etc…), see for example Shneidman ES. Perspectives on suicidology. Further reflections on suicide and psychache. Suicide Life Threat Behav. 1998 Fall; 28(3):245-50; Shneidman, Edwin S. Suicide as psychache: A clinical approach to self-destructive behavior. Jason Aronson, 1993; Orbach I, Mikulincer M, Sirota P, Gilboa-Schechtman E. Mental pain: a multidimensional operationalization and definition. Suicide Life Threat Behav. 2003 Fall; 33(3):219-30.
  3. Line 34-37: I would suggest to also consider other evidences highlighting the link between psychache and suicide, e.g, Verrocchio MC, Carrozzino D, Marchetti D, Andreasson K, Fulcheri M, Bech P. Mental Pain and Suicide: A Systematic Review of the Literature. Front Psychiatry. 2016;7:108. Published 2016 Jun 20. doi:10.3389/fpsyt.2016.00108; Ducasse D, Holden RR, Boyer L, Artéro S, Calati R, Guillaume S, Courtet P, Olié E. Psychological Pain in Suicidality: A Meta-Analysis. J Clin Psychiatry. 2018 May/Jun;79(3):16r10732. doi: 10.4088/JCP.16r10732.
  4. Line 41-44: I would suggest to add references related to the cited cross-sectional studies. Further, limitation of such studies might be related to their design as a cross-sectional study does not allow a causal interpretation.
  5. Line 45-46: with regard to the sentence “the diagnoses historically associated with suicide risk (e.g. personality disorders, substance abuse)” I would suggest to add some evidence/citations for such statement e.g. Too LS, Spittal MJ, Bugeja L, Reifels L, Butterworth P, Pirkis J. The association between mental disorders and suicide: A systematic review and meta-analysis of record linkage studies. J Affect Disord. 2019 Dec 1;259:302-313. doi: 10.1016/j.jad.2019.08.054; Brezo J, Paris J, Turecki G. Personality traits as correlates of suicidal ideation, suicide attempts, and suicide completions: a systematic review. Acta Psychiatr Scand. 2006 Mar;113(3):180-206. doi: 10.1111/j.1600-0447.2005.00702.x.
  6. Line 53: with regard to the sentence “Starting from the assumption that mental pain is a measurable construct” I would suggest to add the potential citation or to clarify whether this is a hypothesis.
  7. Line 55: with regard to this aim “to identify a psychometric tool in the context of suicide prevention”, a power analysis should be encouraged.

Method:

  1. Line 71: study design does not seem cross-sectional but longitudinal. Further, it seems that participants were enrolled during a clinical visit and signed “the informed consent for the anonymous use of data for research purposes”. It sounds like scales were administered for clinical purposes and then collected data would have been anonymously published, but it is not clear if participants received clear information on study design and if they also signed the consent for collecting data during the follow up period. Ethic statement should be added related to the study’s approval.
  2. Line 108-109: I would suggest to better specify-> how hospital (PORTALE) and territorial (PSYCHE) databases works; and how medical records were analyzed.
  3. As stated in the abstract, I would suggest authors to add clearly in the method sections that: Clinical and sociodemographic variables of the sample were collected.
  4. Lines 84-106: The study was performed in Italy. Therefore, I would suggest authors to better describe, as they did with CTQ [ref 17,18], whether the original, translated or Italian version of the questionnaires was used, to add the related references and to specify if these instruments are validated. Further, I would suggest specifying if clinicians were trained to administer some of these scales.

Results:

Line 126: polypharmacotherapy term is too broad, I would suggest to give some details

Line 127: I would suggest authors to elucidate what means most in terms of N, % of subjects for each reported diagnosis.

11 out of 105 subjects committed an AS, this should be clearly stated in the abstract

Discussion:

Overall, the discussion is quite short and would benefit from a more explicit contextualization of what the study adds to extant knowledge. I would suggest authors to briefly describe results at the beginning of the discussion, to better elucidate (adding references) which results confirmed previous data in literature (as used terms are too broad, for example line 178: “this study confirm the existing literature data”; line 180-181: “as reported in literature”, “we found results in line with literature” I would suggest to add references) and to add greater details regarding study's limitations in a specific limitation section: e.g. limited sample size, confounding factors, etc. Further, the follow up period was conducted during the COVID-19 pandemic. Despite authors state that this aspect was not considered as “the purpose of the study is to analyze the internal factor of mental pain rather than external factors”, it should be noticed that evidence highlights increased suicide risk (e.g., Farooq, S., et al., Suicide, self-harm and suicidal ideation during COVID-19: A systematic review. Psychiatry Res, 2021. 306: p. 114228) and increased addictive behaviors (Panno, A., Carbone, G. A., Massullo, C., Farina, B., & Imperatori, C. (2020). COVID-19 Related Distress Is Associated With Alcohol Problems, Social Media and Food Addiction Symptoms: Insights From the Italian Experience During the Lockdown. Front Psychiatry, 11, 577135) during the pandemic

Author Response

Reviewer 2

This manuscript provides preliminary data to the potential use of Beck Depression Inventory (BDI-II) and Mental Pain Questionnaire (OMMP) as screening tools of suicide risk. However, the manuscript would benefit of major changes.

Background: 

  • Line 28-29: with regard to the estimation of victims of suicide I would suggest to consider also World Health Organization. Suicide worldwide in 2019. Global HealthEstimates, JUNE 2021.

Thank you for your comment. We added the reference as suggested.

  • Line 32: I would suggest authors to better describe dimensions related to psychache (e.g. guilt, loneliness, fear, irreversibility of pain, emotional flooding, etc…), see for example Shneidman ES. Perspectives on suicidology. Further reflections on suicide and psychache. Suicide Life Threat Behav. 1998 Fall; 28(3):245-50; Shneidman, Edwin S. Suicide as psychache: A clinical approach to self-destructive behavior. Jason Aronson, 1993; Orbach I, Mikulincer M, Sirota P, Gilboa-Schechtman E. Mental pain: a multidimensional operationalization and definition. Suicide Life ThreatBehav. 2003 Fall; 33(3):219-30.

Thank you for your suggestion. We described dimensions related to psychache with reference to the suggested paper.

  • Line 34-37: I would suggest to also consider other evidences highlighting the link between psychache and suicide, e.g, Verrocchio MC, Carrozzino D, Marchetti D, Andreasson K, Fulcheri M, Bech P. Mental Pain and Suicide: A Systematic Review of the Literature. Front Psychiatry. 2016;7:108. Published 2016 Jun 20. doi:10.3389/fpsyt.2016.00108; Ducasse D, Holden RR, Boyer L, Artéro S, Calati R, Guillaume S, Courtet P, Olié E. Psychological Pain in Suicidality: A Meta-Analysis. J ClinPsychiatry. 2018 May/Jun;79(3):16r10732. doi: 10.4088/JCP.16r10732.

Thank you for your suggestion. We considered other evidences highlighting the link between psychahe and suicide by reference to the suggested article.

  • Line 41-44: I would suggest to add references related to the cited cross-sectional studies. Further, limitation of such studies might be related to their design as a cross-sectional study does not allow a causal interpretation.

Thank you for your observation. We added references related to the cited studies.

  • Line 45-46: with regard to the sentence “the diagnoses historically associated with suicide risk (e.g. personality disorders, substance abuse)” I would suggest to add some evidence/citations for such statement e.g. Too LS, Spittal MJ, Bugeja L, Reifels L, Butterworth P, Pirkis J. The association between mental disorders and suicide: A systematic review and meta-analysis of record linkage studies. J Affect Disord. 2019 Dec 1;259:302-313. doi: 10.1016/j.jad.2019.08.054;Brezo J, Paris J, Turecki G. Personality traits as correlates of suicidal ideation, suicide attempts, and suicide completions: a systematic review. Acta PsychiatrScand. 2006 Mar;113(3):180-206. doi: 10.1111/j.1600-0447.2005.00702.x.

We added some evidences to the mentioned statement as suggested.

  • Line 53: with regard to the sentence “Starting from the assumption that mental pain is a measurable construct” I would suggest to add the potential citation or to clarify whether this is a hypothesis.

We added a citation to the mentioned sentence as suggested.

  • Line 55: with regard to this aim “to identify a psychometric tool in the context of suicide prevention”, a power analysis should be encouraged.

Thank you for your observation. We did not provide a power analysis, but we evaluated the sample size based on previous investigation [8]. We added a sentence in the material and methods section.   

Method: 

  • Line 71: study design does not seem cross-sectional but longitudinal. Further, it seems that participants were enrolled during a clinical visit and signed “the informed consent for the anonymous use of data for research purposes”. It sounds like scales were administered for clinical purposes and then collected data would have been anonymously published, but it is not clear if participants received clear information on study design and if they also signed the consent for collecting data during the follow up period. Ethic statement should be added related to the study’s approval.

Thank you for your observations. We clarified the research design type specifying that is “longitudinal”. Regarding the patients’ information process, the information regarding Ethics Committee or Institutional Review Board approval has been specified in the text. The patients have been signed an informed consent for the use of data anonymously for research purposes. Since data were made anonymous and unidentifiable, the Provincial Health Ethical Review Board (Ethics Committee of Insubria – Azienda Socio Sanitaria Sette Laghi, Varese, Italy) consulted prior to the beginning of the study, has confirmed that, since the study consists in a longitudinal research, it does not need approval process from the Board. 

  • Line 108-109: I would suggest to better specify-> how hospital (PORTALE) and territorial (PSYCHE) databases works; and how medical records were analyzed.

We better specified the functioning of the databases and the records analysis in the text.

  • As stated in the abstract, I would suggest authors to add clearly in the method sections that: Clinical and sociodemographic variables of the sample were collected.

We added in the method section the cited sentence as suggested.

  • Lines 84-106: The study was performed in Italy. Therefore, I would suggest authors to better describe, as they did with CTQ [ref 17,18], whether the original, translated or Italian version of the questionnaires was used, to add the related references and to specify if these instruments are validated. Further, I would suggest specifying if clinicians were trained to administer some of these scales.

Thank you for your observation which gives us the opportunity to specify better this point. The questionnaires used are Italian validated versions. GL performed online training for administration of the tools (Department of Psychiatry, Sapienza University of Rome). Thus, we specified in the text that clinicians were trained to administer the psychometric scales.

Results: 

  • Line 126: polypharmacotherapy term is too broad, I would suggest to give some details

We improved the sentence as suggested.

  • Line 127: I would suggest authors to elucidate what means mostin terms of N, % of subjects for each reported diagnosis.

“No new AS” refers to the group of patients that did not committed new AS in percentage

  • 11 out of 105 subjects committed an AS, this should be clearly stated in the abstract

We clarified the mentioned statement as suggested.

Discussion: 

Overall, the discussion is quite short and would benefit from a more explicit contextualization of what the study adds to extant knowledge. I would suggest authors to briefly describe results at the beginning of the discussion, to better elucidate (adding references) which results confirmed previous data in literature (as used terms are too broad, for example line 178: “this study confirm the existing literature data”; line 180-181: “as reported in literature”, “we found results in line with literature” I would suggest to add references) and to add greater details regarding study's limitations in a specific limitation section: e.g. limited sample size, confounding factors, etc. Further, the follow up period was conducted during the COVID-19 pandemic. Despite authors state that this aspect was not considered as “the purpose of the study is to analyze the internal factor of mental pain rather than external factors”, it should be noticed that evidence highlights increased suicide risk (e.g., Farooq, S., et al., Suicide, self-harm and suicidal ideation during COVID-19: A systematic review. Psychiatry Res, 2021. 306: p. 114228) and increased addictive behaviors (Panno, A., Carbone, G. A., Massullo, C., Farina, B., &Imperatori, C. (2020). COVID-19 Related Distress Is Associated With Alcohol Problems, Social Media and Food Addiction Symptoms: Insights From the Italian Experience During the Lockdown. Front Psychiatry, 11, 577135) during the pandemic.

Thank you for your observations. We modified the discussion as suggested adding more specific references. We added more details in the study’s limitations section. We discussed the evidence of increased suicide risk and addictive behaviors during the COVID-19 pandemic in the text by reference to the suggested papers.

Round 2

Reviewer 2 Report

The manuscript deserves publication